# Preparation and Characterization of Cyclodextrin Coated Red Phosphorus Double−Shell Microcapsules and Its Application in Flame Retardant Polyamide6

**DOI:** 10.3390/polym14194101

**Published:** 2022-09-30

**Authors:** Shangkai Lu, Junbang Liu, Lijuan Zeng, Lianghui Ai, Ping Liu

**Affiliations:** State Key Laboratory of Luminescent Materials and Devices, Research Institute of Materials Science, South China University of Technology, Guangzhou 510640, China

**Keywords:** melamine borate, cyclodextrin, red phosphorus, microcapsules, flame retardant

## Abstract

Using the melamine borate and crosslinked β−cyclodextrin as shell materials, the double−shell microcapsules (Mic−DP) of red phosphorus (RP) was prepared, and its flame−retardant effect on polyamide 6 (PA6) was investigated. Compared with RP, Mic−DP showed lower hygroscopic and better inoxidizability. The limiting oxygen index (LOI) of PA6/13%Mic−DP was 27.8%, and PA6/13%Mic−DP reached V−0 rating. After the addition of 13% Mic−DP, the total exothermic (THR), peak exothermic (PK−HRR), and average effective thermal combustion (AV−EHC) rates of PA6 decreased. In addition, in order to investigate its flame−retardant mechanism, the pyrolysis gas chromatography−mass spectrometry (Py−GC−MS), scanning electron microscopy (SEM), and Fourier transform infrared (FT−IR) methods were used, and the results showed that mic DP acted as a flame retardant in the gas and condensed phases. The Mic−DP exhibited good compatibility and dispersibility in PA6.

## 1. Introduction

Polyamide 6 (PA6), also known as nylon 6, is a common engineering plastic used in electronics and communication industries, due to its oil resistance, electrical insulation, and good mechanical properties [1,2]. PA6 often works at high temperature, humidity, or voltage, in which it is easy to cause fire. Therefore, flame retardants have been used to improve the flame retardance of PA6 [3,4,5].

Halogen−based flame retardants have high flame−retardant efficiency, but they easily produce large amounts of smoke and toxic corrosive gases during the burning [6,7,8]. Therefore, non−halogenated flame retardants attracted more attention [9,10,11]. Red phosphorus (RP) is a common and practical phosphorus−based flame retardant. As an elemental substance, RP has high phosphorus weight ratio for meeting high flame−retardant efficiency [12,13]. However, RP is exposed to hygroscopic oxidation in air and has poor compatibility with polymers, which limits its application. To overcome these disadvantages, microencapsulated RP has been performed [14,15]. Wu et al. [16] prepared RP microcapsules by forming a protective layer on the surface of RP with melamine formaldehyde. Its thermal stability and water resistance were improved. It was also applied to LLDPE, and the LOI value reached 32.0% at the addition amount of 10%. The protective layer is formed on the surface of RP to fabricate a core−shell structure, thus reducing its direct contact with the environment. Chen et al. [17] prepared RP microencapsulated flame retardants (RP@CH/LS) using the electrostatic interaction of lignosulfonate (LS) and chitosan (CH) to form a protective layer on the surface of red phosphorus. It was also applied to epoxy resin, and its LOI value was improved to 30.6% and reached V−0 grade at the addition amount of 7%.

With the increasing attention to environmental protection, bio−based flame retardants as green and renewable flame retardants have received the attention of researchers. β−cyclodextrin (β−CD), which is composed of multiple glucose units and exhibits a hollow truncated cone arranged structure, is an oligosaccharide produced by the enzymatic hydrolysis of starch [18,19]. The structure contains multiple hydroxyl groups and has a good thermal stability and char−forming ability [20,21]. Ding et al. [22] prepared ammonium polyphosphate (APP) microcapsules (MCAPP) by coating them with HDI and β−cyclodextrin. It was found that MCAPP showed reduced hygroscopicity and better compatibility when it was added to PP, and its LOI value increased to 30.9% and reached V−0 rating at 25% addition.

In order to overcome the defects of RP and improve the flame−retardant efficiency of RP, the double−shell microcapsules (Mic−DP) of red phosphorus (RP) was prepared, which used the melamine borate and crosslinked β−cyclodextrin as shell materials, and its flame retardant on polyamide 6 (PA6) was investigated.

## 2. Materials and Methods

### 2.1. Material

Polyamide 6 (PA6, TP4208) was purchased from Zig Sheng Industrial Co., Taiwan, China. Red phosphorus (RP, 70 wt% content of phosphorus) was supplied by Guangzhou YS Flame Retardant Materials Co., Ltd., Guangzhou, China. Tween−80 and β−cyclodextrin (β−CD) were purchased from Beijing HWRK Chem Co., Ltd., Beijing, China. Toluene diisocyanate (TDI) was purchased from Wanhua Chemical Group Co. Ltd., Yantai, China. Pyridine and acetone were purchased from Guangzhou Chemical Reagent Factory, Guangzhou, China. Microcapsules based on melamine borate coated red phosphorus (Mic−MBP) were prepared by our laboratory [23].

### 2.2. Characterization

Fourier transform infrared spectroscopy (FTIR) was recorded on a Vector 33−MIR FTIR spectrometer using KBr pellets. Thermogravimetry (TG) analysis was conducted with a TG209F3 thermal analyzer; the heating rate was 20 °C/min, and the nitrogen flow rate was 40 mL/min. The TG analysis software was Proteus Analysis. The morphology of RP, Mic−DP, and char residues was measured on an EVO18 scanning electron microscopy (SEM). Elemental analysis of the residual char was tested using an EVO18 energy dispersive X−ray. X−ray diffraction (XRD) was analyzed by an X’Pert Pro X−ray diffractometer. Laser Raman spectroscopy (LRS) was performed by using a laser Raman spectrometer (Renishawin Via, Renishaw). PY−GC−MS was measured by a GCMSQP 2010, plus pyrolysis−gas chromatography mass spectrometer. The pyrolysis temperature was 700 °C, and the heating rate was 10 °C/ms; the silane capillary column was used RXI−1 from RESTEK.

To investigate the moisture absorption of RP and Mic−DP, 1 g of RP or Mic−DP was placed in watch glass, into a container with a relative humidity at 75%. To increase the accuracy of the experimental results, the percentage change in weight after one week was calculated by weighting. To study the oxidation resistance of RP and Mic−DP, 10 g RP or Mic−DP were added to 200 mL deionized water in a round flask and stirred for one hour. A total of 50 mL of the filtrate was then titrated with 0.1 mol/L NaOH to calculate NaOH consumption. Equation (1) is the calculation formula of NaOH consumption.
W_NaOH_ = (40 × C × V)/(G × 50/200 × 1 h)(1)
where V is the volume of NaOH solution consumed, G is the mass of sample (10 g), C is the concentration of NaOH (0.1 mol/L), and 1 h is one hour.

The limiting oxygen index (LOI) was conducted with a FTT oxygen index tester, according to the ASTM D2863, the size of tested samples were 80 × 10 × 4 mm^3^. Vertical burning tests were conducted with a FTT vertical burning tester, according on the ASTM D2863, the size of tested samples were 125 × 12.7 × 3.2 mm^3^. Cone calorimeter tests were conducted with a FTT cone calorimeter, and the incident flux was 50 kW/m^2^. According to the ISO−5660, the size of tested samples were 100 × 100 × 3 mm^3^. The impact strengths were conducted with a ZCJ 1320 impact testing machine. According to the GB/T 21189−2007, the size of tested samples was 80 × 10 × 4 mm^3^.

### 2.3. Preparation of Mic−DP

The preparation route of Mic−DP was shown in Figure 1. Then the Mic−MBP was placed in an oven at 75 °C for 3 h. β−CD (3.1 g); a total of 60 mL pyridine were added into a three−necked flask, and then stirred at 80 °C until the solution clarification. A total of 10 g dried Mic−MBP and 0.1 g Tween−80 were added into the solution and stirred for 30 min to obtain a uniform dispersion. A total of 1.9 g TDI was dissolved in 5 mL pyridine, added dropwise into the mixture in 30 min, and then further reacted for 5 h at 80 °C. After that, 120 mL acetone were added and stirred for 30 min. The target products were washed with acetone and filtered by Brinell Funnel, and then grinded to powder after oven dried at 100 °C for 24 h [24].

### 2.4. Preparation of PA6 Samples

PA6 and Mic−DP were dried in oven at 100 °C for 6 h before processing. Table 1 lists the composition of PA6 samples. PA6 and Mic−DP were melt−mixed in a rheometer at 220 °C for 10 min. Then, the samples were hot−pressed at 220 °C for 5 min into sheet. After that, the sheet was cut into different size for testing.

## 3. Results and Discussion

### 3.1. Characterization of RP and Mic−DP

Figure 1 shows the FT−IR spectra of RP and Mic−DP. The absorption peaks of Mic−DP, including 2926 cm^−1^ (−CH−), 1701 cm^−1^ (C=O), 1600 cm^−1^ (−NH), 1141 cm^−1^ (−C−O−C), and 1030 cm^−1^ (C−O), were evidenced. In addition, the absorption band of −NCO at 2270 cm^−1^ did not appear, which indicated the β−CD was crossed with TDI and Mic−DP.

The SEM images of RP, Mic−MBP, and Mic−DP are shown in Figure 2. The surface of RP was smooth and had sharp edges and corners. As seen in Figure 2b,c, the microscopic surfaces of Mic−MBP and Mic−DP showed a significant change. After encapsulation, the surface of Mic−DP became rough, due to the shell of crossed β−CD. Additionally, as the number of cladding layers increased, the agglomeration of RP gradually decreased.

Table 2 shows the surface elemental composition of RP, Mic−MBP, and Mic−DP. It could be seen that the P content of Mic−MBP and Mic−DP declined, while the O and N content increased, compared to RP. This is due to the high elemental content of oxygen and nitrogen in the melamine borates and β−CD cladding. Compared to Mic−MBP, Mic−DP had an increased nitrogen element, due to the higher number of hydroxyl groups in β−CD. This is an indication that the desired product, Mic−DP, was obtained.

Table 3 presents the hygroscopicity and oxidation resistance of RP and Mic−DP. The moisture absorption ratio and inoxidizability of RP was 6.19% and 12.64 mg NaOH/g·h, respectively. After encapsulation, the value of them declined apparently to 0.74% and 0.40 mg NaOH/g·h. It was indicated that microencapsulation could reduce RP’s reaction with water to form acid.

The TG and DTG curves of RP and Mic−DP are shown in Figure 3. The first and second degradation steps of Mic−DP are at 279 and 317 °C, due to the decomposition of the shell of Mic−DP, including the decomposition of melamine and crossed β−CD [25]. The third degradation step was at 518 °C, which was due to RP sublimation, forming white phosphorus. The residues of Mic−DP and RP at 800 °C were 14.4% and 5.3%, respectively.

### 3.2. LOI and UL−94 Vertical Burning of PA6 Samples

The results of the LOI and UL−94 of PA6 samples are listed in Table 4. The LOI value of PA6 was only 21.8%, and PA6 could not be rated, due to the severe dripping and poor self−extinguishing ability. With the addition of 13 wt% RP, the LOI value of PA6/13%RP increased to 27.5%, and PA6/13%RP merely reached V−2 because of dripping. However, when 13 wt% Mic−DP was incorporated in PA6, PA6/13%Mic−DP reached V−0 with no dripping and self−extinguishing in a short time. In addition, the LOI value of PA6/13%Mic−DP also increased to 27.8%. With less RP content, PA6/Mic−DP had better flame retardancy. Mic−DP decomposed to form H_2_O and inert gas containing nitrogen, thus reducing the concentration of flammable gases. RP turned into polyphosphoric acid through thermal oxidation during combustion and reacted with crossed β−CD, which promoted the formation of carbon layer and improved anti−dripping performance of PA6.

### 3.3. TG and DTG of PA6 Samples

The TGA and DTG curves of PA6 samples are illustrated in Figure 4, and the correlative experiment data are listed in Table 5. The PA6 samples displayed single step pyrolysis. With the incorporation of RP, the decomposition onset temperature, and maximum decomposition rate temperature of PA6 sample decreased, due to the formation of phosphoric acid or phosphoric anhydride by thermal oxidation, which promoted the decomposition of PA6. For the Mic−DP, the decomposition onset temperature of PA6/Mic−DP was slightly lower than PA6/RP, due to the decomposition of the shell materials. However, the maximum decomposition rate temperature of PA6/Mic−DP was higher than that of PA6/RP, thus indicating that Mic−DP promoted carbonization and delayed decomposition of PA6, which compared to PA6/RP.

### 3.4. Cone Calorimeter Data of PA6 Samples

The cone calorimeter test has important reference value in evaluating the performance and design of flame−retardant materials and fire prevention. Figure 5 exhibits the heat release rate (HRR), total heat release (THR), and mass loss curves of the PA6 samples. Table 6 lists the relevant test data.

The peak HRR (pk−HRR) and THR of PA6 were 869.5 kW/m^2^ and 96.7 MJ/m^2^, respectively. With incorporation of Mic−DP and RP, the pk−HRR and THR of PA6 samples had decreased. The pk−HRR and THR of PA6/13%Mic−DP were 319.7 kW/m^2^ and 62.7 MJ/m^2^, which decreased by 63.2% and 35.1%, compared with PA6. As for PA6/13%RP, the value of pk−HRR was higher than PA6/13%Mic−DP. With the addition of Mic−DP, RP reacted with crossed β−CD, thus promoting the carbonization of polymer, which retarded the heat transfer and isolated oxygen from polymer. The results demonstrated that PA6/Mic−DP had good flame retardancy.

The burning intensity of the volatiles in the gas phase of the sample can be reflected by the average effective combustion heat (av−EHC). During heating, PA6 is decomposed to form flammable gases and small molecule fragments, and it combusts vigorously with the oxygen in the air, thus resulting in the highest av−EHC production. In the gas phase, the phosphorus oxygen radical formed by RP decomposition, which traps the reactive radicals of the combustion reaction, interrupts the chain reaction of combustion, and reduces the combustion intensity of the gas phase. Therefore, the av−EHC of PA6/RP decreased. In addition, the non−combustible gases decomposed by the shell material of Mic−DP reduce the concentration of volatile gases. On the other hand, the carbon layer formed by PA6/Mic DP also inhibits the release of combustible volatiles, which leads to the decrease of av−EHC.

### 3.5. Residual Char from Cone Calorimeter Tests

Figure 6 displays the images of residual chars from PA6 samples. The PA6 decomposed completely, almost without any residual char, thus resulting in the poor flame retardancy. However, with the incorporation of Mic−DP, PA6/Mic−DP formed an expanded carbon layer to cover the surface of samples after ignited, which blocked the transfer of heat and oxygen efficiently. When the amount of Mic−DP grew, the residual chars of PA6/Mic−DP were increased and became more continuous. Whereas the residual chars of PA6/13%RP was lower than PA6/13%Mic−DP, and the continuity of carbon layer was also worse than PA6/13%Mic−DP. For the PA6/13%RP, RP reacted with PA6 in thermal oxidation directly to form phosphoric acid and phosphorous free radicals during combustion. Phosphoric acid promoted the carbonation of polymer and phosphorous free radicals trapped the active free radicals to reduce combustion intensity of gas phase. However, the carbon layer of PA6/13%RP had poor continuity and did not expand without carbon and gas sources.

The SEM images of residual chars from PA6/13%Mic−DP and PA6/13%RP were shown in Figure 7. For the PA6/13%RP, the residual carbon was more loose, with more holes. For the PA6/13% Mic−DP, the residual chars were more continuous and denser, due to the shell material of Mic−DP, which decomposed from nonflammable gas and provided the carbon source.

Generally, the graphitic structure of the residual chars can be investigated by Raman spectroscopy. The Raman spectra of the residual chars of PA6/13%Mic−DP and PA6/13%RP were shown in Figure 8. The D and G bands appeared at 1364 and 1614 cm^−1^, which represent the two structures of disordered graphite and graphite, respectively. This is related to the vibration of carbon atoms in graphite layers. The integral ratio of the band area (I_D_/I_G_) reflected the graphitization degree of the residue chars. The I_D_/I_G_ of PA6/13%Mic−DP (2.3) was lower than PA6/13%RP (2.44), indicating the higher graphitization degree of the residual char of PA6/13%Mic−DP, which demonstrated that the carbon layer of PA6/13%Mic−DP was more regular and continuous.

### 3.6. Morphologies of PA6 Composites at Different Temperatures

The photographs of PA6 samples stored in a muffle furnace at different temperatures for 15 min are shown as Figure 9. The PA6 became discolored and transparent at 250 °C, and the sample morphology was severely damaged at 400 °C. When the temperature rose to 600 °C, PA6 decomposed completely without any residual. PA6/13%Mic−DP expanded at 450 °C and formed an expanding carbon layer at 650 °C. While the morphology of PA6/13%RP was damaged at 400 °C, it changed to a viscous state from the solid state. RP reacted with PA6 during combustion oxidizing to various phosphoric acid derivatives. However, without carbon and gas sources, the structure of carbon layer was irregular, which reduced the thermal stability of polymer.

After being placed in a muffle furnace at 650 °C for 15 min, the FTIR spectra of residual char from PA6/13%RP and PA6/13%Mic−DP are shown in Figure 10. The characteristic peaks in PA6/13%Mic−DP spectra were similar to the PA6/13%RP. The absorption peaks at 1165 and 1076 cm^−1^ corresponded to the characteristic absorption peak of P−O−C. The absorption peak located at 939 cm^−1^ was associated with the characteristic absorption peak of P−O−P. The absorption peak at 1645 cm^−1^ corresponded to the characteristic absorption peak of C=C. The results indicated PA6/13%Mic−DP and PA6/13%RP decomposed to form phosphoric acid derivatives, thus promoting the carbonation of PA6. In addition, with the crossed β−CD, it formed more stable structures containing P−O−P chemical bonds and was effectively promoted to char in combustion.

### 3.7. Pyrolysis–Gas Chromatography–Mass Spectrometry

The Figure 11 shows the PY–GC–MS results of PA6 and PA6/13%Mic−DP at 700 °C. Table 7 and Table 8 list the assignment of peaks in the mass spectrum. PA6 decomposed at high temperature to form a wide variety of volatiles, and they presented high intensities in the mass spectra, due to the active free radicals H· and ·OH [26]. The active free radicals were formed by the decomposition of PA6, which promoted the chain breaking decomposition of the polymer. When Mic−DP was added, the intensities of gaseous volatiles decomposed by PA6 samples decreased. Mic−DP consumed active free radicals, thus reducing the polymer—free radical reaction rate and delaying the decomposition of PA6. So, it played a flame−retardant role in the gas phase.

### 3.8. Flame−Retardant Mechanism

The shell–core structure of Mic−DP made it a small, self−contained, intumescent, flame−retardant system. The RP acted as acid source, and the shell materials acted as gas and carbon sources. During combustion, Mic−DP first decomposed to nonflammable inert gases, including NH_3_ and H_2_O [23], resulting in the reduction of concentration of flammable volatiles. Then, RP formed phosphorous acid and phosphorous free radicals, which reacted with crossed β−CD to promote the formation of carbon layer and trapped the active free radicals in the gas phase, respectively. The formation of carbon layer insulated oxygen effectively, and the reduction of free radicals led to the interruption of the chain reaction. Therefore, Mic−DP played a flame−retardant role in gas and condensed phases.

### 3.9. Mechanical Property

Table 9 displays the mechanical properties of the PA6 samples. With the addition of 13% RP, the impact strength of PA6 sample declined from 12.9 to 2.7 kJ·m^−2^, due to the poor compatibility of RP. After encapsulation by organic compounds, the compatibility of Mic−DP had been improved and the impact strength of PA6/13%Mic−DP was 8.4 kJ·m^−2^. Mic−DP exhibited good compatibility and dispersibility in PA6. The results indicated Mic−DP not only improved the flame−retardant property of PA6, but also reduced the effect on its mechanical properties.

## 4. Conclusions

In this work, β−cyclodextrin−coated red phosphorus double−shell microcapsules (Mic−DP) were prepared. Compared with RP, the moisture absorption rate of Mic−DP decreased, and its antioxidant capacity was enhanced. After adding 13%Mic−DP, the LOI value of PA6/13% Mic−DP reached 27.8% and a V−0 rating in the UL 94 test. The pk−HRR of PA6/13% Mic−DP decreased from 869.5 to 319.7 kW/m^2^, THR decreased from 96.7 to 62.7 MJ/m^2^, and av−EHC decreased from 21.1 to 12.4 MJ/kg, compared with PA. Compared with PA6/13% RP, the impact strength of PA6/13% mic−DP was increased to three times that of PA6/13%. The microencapsulated flame retardant exhibited its good flame−retardant efficiency, while reducing its effect on the mechanical properties of the material.

## Data Availability

The authors declare that all data supporting the results of this study are available in the manuscript.

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
