# Peer review of "Preparation and Characterization of Cyclodextrin Coated Red Phosphorus Double−Shell Microcapsules and Its Application in Flame Retardant Polyamide6"

_polymers, 2022, doi:10.3390/polym14194101_

Round 1

Reviewer 1 Report

Reviewer:

Comments to the Author

Title: Preparation and characterization of cyclodextrin coated red phosphorus double-shell microcapsules and its application in flame retardant polyamide6

1. Abstract

-          There is punctuation mistake in the first sentence.

2. Introduction

-          There is punctuation mistake in the last paragraph.

3. Materials and Methods

-          In the subsection 2.3. It is not clear this method new or well-known. If it is well-known the corresponding references should be presented.

4. Results and discussion

-          There should be a separate subsection in the results and discussion section. This subsection should provide information about the reaction and its mechanism.

Reviewer 2 Report

This manuscript deals with the preparation and characterization of cyclodextrin-coated red phosphorus double-shell microcapsules and their possible effect on the flame retardancy of polyamide 6. The introduction should provide more information and introduce the reader to the issue. The methodology is not sufficient, some techniques are not described at all.

A few comments are added:

1)      The introduction should provide more information on the given topic.

2)      Pyrolysis Gas Chromatography-Mass Spectrometry is not described at all. The authors should add the interval during which the temperature pulse was applied to the material.

3)      The authors should add the description of the GC-MS column used, the temperature gradient, the MS spectra library, and the probability of the MS spectra evaluated for the structure with the MS library.

4)      The authors should properly describe the filter they used.

5)      The authors should add the water content of all chemicals used and discuss the effect of water content or air humidity on the reaction with TDI.

6)      Figure 2 does not appear to be representative and no conclusion can be done about the shell of crossed β-CD. The authors should delete this Figure or discuss this Figure more appropriately.

7)      The authors should discuss the decrease in the size of Mic-DP observed in Figure 2.

8)      The authors should provide more sufficient confirmation for the proposed double-shell microencapsulation.

9)      The author should correct the results from elemental analysis for RP or correct the sample name. Red phosphorus cannot contain carbon and oxygen.

10)   The authors should use RP only for red phosphorus and not for other substances.

11)   The authors should provide an experiment or cite literature for active free radicals for PA6 statement.

Reviewer 3 Report

The manuscript entitled "Preparation and characterization of cyclodextrin coated red phosphorus double-shell microcapsules and its application in flame retardant polyamide6" by Ping Liu * , Junbang Liu , Shangkai Lu , Lianghui Ai , Lijuan Zeng contains a considerable amount of experimental data obtained by various techniques. 

However, the presentation, interpretation and discussion of the experimental material need consistent improvement to make the article publishable. The attached, reviewed form of the manuscript contains over 60 comments/observations/suggestions.

There are wrong and/or conflicting explanations/interpretations of experiments.

A thorough and extensive English editing of the whole manuscript is needed. 

References are not utilized throughout the discussion section, i.e. they are not referred at. Some important, closely related references are missing. 

The authors should consider the above observations and engage in a sustained effort to support their own data.

Reviewer 4 Report

The manuscript (polymers-1856095) presents the utilization of encapsulated red-phosphorous as flame retardant in polyamide 6. The authors addressed the effect on the mechanical properties and flame retardancy and mechanism for this action. The manuscript is logically arranged and well structured. However, I have the following observations prior to publication:

1)      the actual structuring of the double walled red-phosphorous capsules is not accurately presented in the manuscript. There is only a reference to the author’s previous paper. For clarity the experimental procedures should be expanded.

2)      Scheme 1 should be improved presenting the functional groups present at the surface of the Mic-MBP.

3)      Why do the authors use beta-cyclodextrines for the second shell? The choice should be explained in the manuscript.

4)      Some English improvement is required: minor editing errors such as extra “.” (see page 1 - β-cyclodextrin as shell materials, the double-shell microcapsules (Mic-DP) of red phosphorus (RP) was prepared. and its flame retardant on polyamide 6 (PA6) was investigated.) Also, phrasing should be improved: page 7 - “The PA6 decomposed completely with almost any residual char,” – should be “almost without any”.

5)      Novelty and implications applicability of the results should be better highlighted and presented.

Round 2

Reviewer 3 Report

The revised version of this manuscript addressed most of the issues/recommendations/suggestions generated by the first version submitted. 

A "clean" form of the manuscript (without the tracked changes) would have been more suitable for a fresh reading. I therefore would recommend a careful editorial check for typos of the whole manuscript.

Author Response

Thank you for your valuable comment.We have been checked the whole manuscript for typos carefully.